# Dehydrated Sauerkraut Juice in Bread and Meat Applications and Bioaccessibility of Total Phenol Compounds after In Vitro Gastrointestinal Digestion

**Liene Jansone \*, Zanda Kruma** **, Kristine Majore and Solvita Kampuse**

Department of Food Technology, Faculty of Food Technology, Latvia University of Life Sciences and Technologies, Rigas iela 22, LV-3004 Jelgava, Latvia
\* Correspondence: liene.jansone@llu.lv

**Abstract:** The aim of this study was to evaluate dehydrated sauerkraut juice (DSJ) in bread and meat applications and investigate bioaccessibility (BAC) of TPC in the analyzed products. In current research, sauerkraut juice, dehydrated sauerkraut juice, and bread and meat products prepared with dehydrated sauerkraut juice were analyzed. For all of the samples, total phenol content, antiradical activity by ABTS$^+$, bioaccessibility, and volatile compound profile were determined. Additionally, sensory evaluation was performed to evaluate the degree of liking bread and meat with dehydrated sauerkraut juice. The addition of DSJ increased TPC in bread and meat samples. The bioaccessibility was higher for the control samples compared to DSJ samples. It exceeded 1 and is considered as good. DSJ did not promote bioaccessibility. Benzaldehyde was the highest peak area for the Bread DSJ and Meat DSJ samples, giving a roasted peanut and almond aroma. There were no significant differences in degree of liking for structure, taste, and aroma between the control bread and the Bread DSJ, while Meat DSJ was more preferable in sensory evaluation. DSJ could be used in food applications, but further research is necessary.

**Keywords:** fermented cabbage; spray-drying; in vitro digestion; food

## 1. Introduction

With a rising food crisis worldwide, it is crucial to exploit the maximum of the grown crop including a wise utilization of waste or by-products. Fermented cabbage juice is considered an industrial by-product of the technological process of sauerkraut. Sauerkraut juice (SJ) contains vast bioactive complexes such as phenolic compounds and glucosinolates [1] organic acids, sugars and biogenic amines [2,3], vitamins, especially vitamin C, and minerals [4,5]. Research into new technologies on how to exploit this valuable by-product was carried out. The dehydrated sauerkraut juice (DSJ) was obtained by spray-drying [6], and remains rich in minerals, potassium, and calcium being the most abundant, but also iron and magnesium, as well as vitamin C [7]. Physical and chemical composition of SJ and DSJ is described in authors' previous studies [4,6]. To the best of our knowledge, there is scarce research done on a similar product, so all of the presented results are based on authors' previous studies [6,7].

To provide a reliable knowledge for industrial production, investigations about DSJ application in various products is necessary.

In our previous study, DSJ was evaluated in olive oil and sour cream as a salt alternative and compared with the control samples with NaCl. Sauerkraut juice and dehydrated sauerkraut juice contains sugars such as glucose and fructose and also 10–12% NaCl, that is used to ensure the fermentation process. In the sensory evaluation of the samples, tastes such as sweet, sour, salty, garlic, yogurt, and mayonnaise were mentioned in the DSJ samples and overall liking [7]. Therefore, further research was conducted into staple foods such as bread and meat. There are numerous studies on the smart utilization of

by-products simultaneously enhancing the functional properties of food products. The beneficial effect of various plant-based additives in bread and meat applications have been investigated before, such as lyophilized kale powder [8], Moringa peregrina seed husk in wheat bread [9], melon peel juice powder [10], enriched wheat bread with quinoa leaves powder [11], and with mushrooms [12], quinoa flour, nettle leaves [13], green coffee, berries, lyophilized pomegranate peel in minced beef [14], and many others.

As the cabbage and sauerkraut has a very specific aroma, it also remains in the obtained dehydrated sauerkraut juice (DSJ) and sequentially in the food applications. Volatile compounds deliver not only aroma or odor, but also take part in health promoting, for example with antibacterial, antioxidant, and antifungal attributes that metabolize from precursors such as fatty acids, carotenoids, leucine, etc. [15]. For example, thioglucosidase, which is present in Brassicaceae vegetables, acts as anticancer substance.

Considering the health attributes of fermented products, in vitro tests were carried out to determine the bioaccessibility of sauerkraut juice, DSJ, and the products made with the addition of DSJ. To investigate the DSJ in vitro, common foods such as wheat bread and minced meat have been chosen. Wheat bread, with a well-known and neutral taste, is one of the staple foods in most parts of the world [16]. Enrichment of such foods with functional properties would gain the most benefit because of the vast consumption [17].

However, plant-based bioactive compounds, in their prime state, can be easily degraded under environmental influence or can be inconstant under severe medium of the stomach. Spray-drying with a wall material encapsulates the core material, thus preventing its bioactive compounds from premature degradation, sustaining and handling the release, etc. [14,18]. For example, the release of polyphenols of encapsulated fruit juices, had a positive effect in simulated GIT [18].

Determining the bioaccessibility (BAC) of the active compounds can be achieved by simulating the gastrointestinal tract (GIT) to assess their release from a substance and availability for absorption [19,20]. BAC is a comparison of compound content before and after GIT [21]. GIT or in vitro methods are swift, cost and labor effective, and exclude ethical restrictions [19] while ensuring in vivo—a living organism condition–including oral, gastric, and intestinal phases, considering enzymes, pH, time, and bile salts as some of the main factors [20].

Research proves that DSJ is a nutritionally valuable product, therefore a study into its application in food products and in vitro tests to assess its health benefits as a potential functional ingredient would be useful.

The aim of this study was to evaluate dehydrated sauerkraut juice (DSJ) in bread and meat applications and investigate bioaccessibility (BAC) of TPC in the analyzed products.

## 2. Materials and Methods

### 2.1. Sauerkraut Juice and Dehydrated Sauerkraut Juice

Sauerkraut juice, for the analyses and spray-drying, was obtained from the production plant Ltd. "Dimdiņi" (Lizums, Latvia), which ferments cabbage by the traditional recipe. Cabbage is shredded and mixed with NaCl, grated carrots and caraway, pressed, and left to ferment for 14 days. Dehydrated sauerkraut juice was obtained in a vertical Mini Spray-dryer Buchi 290 (Buchi, Flawil, Switzerland), with a starch solution as a carrying agent [6]. For the starch solution, starch (Pure, soluble starch, 162.10 g $mol^{-1}$, CHEMPUR) and deionized water (1:20, accordingly) was heated for 30 min at 90 °C and cooled overnight. It was then mixed with the sauerkraut juice, with the core-to-wall ratio 1:1.5, and the mixture was constantly stirred on a magnetic stirrer while spray drying. The dehydrated sauerkraut juice was kept in two Ziplock bags to be used for further experiments and analyses.

For the sauerkraut juice (SJ) and dehydrated sauerkraut juice (DSJ), total phenol content (TPC–using Folin–Ciocalteu reagent, described below), antiradical activity by $ABTS^+$ assay, vitamin C by iodometric titration, and salt (NaCl) were determined by Mohr's method, all described in previous studies [4,6].

*2.2. Product Preparation*

2.2.1. Bread

The bread samples were made according to the following formulation: for the control sample 1% of NaCl, 2% sugar, 3% yeast, and 60% water ads to the necessary amount (100%) of wheat flour. The ingredients for the dough were placed in a spiral-type dough mixer KM400 (Kenwood Havant, Hampshire, UK) and kneaded for 7 min, let to rest for 10 min, formed in a baguette-like loaf, and proofed for 45 min in a Sveba Dahlen proofing cabinet (Sveba Dahlen AB, Fristadt, Sweden) at $35 \pm 5\,°C$ and 80% humidity. The loaves were then baked in a preheated rotary oven (Sveba Dahlen, Fristad, Sweden) at $200 \pm 3\,°C$ for 17 min, and then cooled down to room temperature for further analysis. For the sample with DSJ–1%, NaCl and 2% of sugar were substituted with 9% DSJ. The DSJ used in the sample, was calculated to obtain the necessary amount of NaCl and sugar, according to the formulation of the control sample. Due to our previous studies, salt content in DSJ used in this experiment is $11\ g\ 100\ g^{-1}$. For the bread formulation, salt equivalence was calculated as 2 g of salt in the control sample, multiplied by 100 g, and divided by 11 g of salt in 100 g DSJ, according to formulation:

$$\text{Salt equivalent} = \frac{2 \times 100}{11}$$

For the Bread DSJ sample, 1000 g flour, 180 g DSJ, 60 g yeast, and 1100 g water were used.

2.2.2. Meat Samples–Minced Pork Sausage

Minced meat samples were prepared as follows: for the control sample blank minced pork was used, obtained from the local shop; for the DSJ90 g of DSJ was added to a 1000 g minced meat. The prepared meat was formed into long sausage-like loafs. Baked in the preheated rotary oven (Sveba Dahlen, Fristad, Sweden) at $200 \pm 3\,°C$ for 15 min.

2.2.3. Samples and Abbreviations

For further use in the text, abbreviations for the samples were made and are described in Table 1.

**Table 1.** The list of samples and abbreviations.

| Sample | Abbreviation |
|---|---|
| Sauerkraut juice | SJ |
| Dehydrated sauerkraut juice | DSJ |
| Bread control sample | Bread C |
| Bread with dehydrated sauerkraut juice | Bread DSJ |
| Meat control sample | Meat C |
| Meat with dehydrated sauerkraut juice | Meat DSJ |

*2.3. Analytical Methods*

2.3.1. Total Phenol Content and Antiradical Activity

For the determination of total phenol content (TPC), a Folin–Ciocalteu reagent was used [22], and TPC was calculated as gallic acid equivalent (GAE) $mg\ 100\ g^{-1}$. To extract TPC, 10 mL of sauerkraut juice, 0.5 g of DSJ, 5 g of GIT outcome from juice and DSJ, and 3 g of bread and meat GIT outcome/content were used. Samples were prepared by magnetically stirring with 20 mL ethanol: water (80:20) for 2 h, then filtered. Next, 0.5 g of DSJ was stirred with 20 mL of deionized water. To determine TPC absorption in the obtained filtrates, a spectrophotometer JENWAY 6300 (Baroworld Scientific Ltd., Staffordshire, UK) was used. The ABTS$^+$ antiradical activity was assessed by a method described by Rokayya [23] with slight modifications. Then, 0.05 mL sample extract was

added to 5 mL ABTS$^+$ solution and absorbance readings were measured at 734 nm after ten minutes of initial mixing.

### 2.3.2. Determination of Volatile Compounds

A method described by Galoburda [24] was used to determine volatile compounds in the raw material (DSJ and sauerkraut juice) and the experimental samples–bread and meat. First, 0.5 g of DSJ and 5 g of sauerkraut juice, 5 g of control and experimental bread and meat samples were placed in a 20 mL glass vial. They were heated and stirred for 10 min at $35 \pm 2$ °C to equilibrate headspace and 30 min with CarboxenTM/Polydimethylsiloxane (CAR/PDMS) fiber (Supelco Inc., Bellefonte, PA, USA) to extract volatiles. Solid-phase microextraction (SPME) techinque was used. The obtained compound data was identified using mass spectral library Nist98.

### 2.3.3. Sensory Evaluation

For the sensory evaluation of bread and meat samples, 52 consumers, based on the complexity/simplicity of the tasting samples [25] and who were willing to try the products, were invited to evaluate the overall liking, structure, taste, and aroma of the bread and meat samples. A 5-point hedonic scale for consumer preferences was used (1–dislike very much and 5–like extremely). Of the customers, 38% were 25 years old or older, and others were 18 to 25 years old. Seventeen percent of all the consumers were male. The tasting samples were prepared/cooked as explained above. Samples were cut into bite-size pieces (to fit easily in a mouth with one bite), and drinking water to cleanse the palate was available at consumers' disposal.

### 2.4. Static In Vitro Digestion Method

For digestibility of sauerkraut juice, DSJ, and experimental bread and meat samples, a standardized static in vitro digestion method by Minekus [20] has been used. It was performed in a model environment of the gastrointestinal tract (GIT)—a bioreactor Multifors 2 (Infors-HT, Bottmingen, Basel, Switzerland) in which digestibility and transit of nutrients are simulated. The process is controlled by the computer program Iris 6 Pallalel bioprocess Control Software (Infors-HT, Bottmingen, Basel, Switzerland). First, 30 g of the sauerkraut juice, bread and meat samples, and 1 g of DSJ were placed in the bioreactor with a pH and temperature control, and simulated saliva fluid (SSF) is then added and kept for 2 min at 37 °C. The transition to the stomach is simulated by introducing the simulated gastric fluid (SGF), which consists of a concentrated electrolyte solution, the enzyme pepsin, $CaCl_2$, and deionized $H_2O$. Gastric acid secretion was simulated by adding 1 M HCl and adjusting pH to $3.0 \pm 0.2$. Digestibility in the stomach is simulated for 2 h. Next, the stomach content is neutralized to pH $7.0 \pm 0.2$ by adding 1 M $NaHCO_3$ and simulated intestinal fluid (SIF), which consists of a concentrated electrolyte solution, enzymes (trypsin, chymotrypsin, α-amylase, lipase), bile salts, $CaCl_2$, and deionized $H_2O$, thus simulating the transit to the duodenum. Digestibility in the small intestine was simulated for 2 h. After the digestion process, the content is frozen to stop the enzymatic activity.

The bioavailability index (BAC) was calculated using the following equation [26]:

$$BAC = \frac{CGE}{CBE}$$

where CGE—the TPC and antiradical activity after gastrointestinal digestion; and CBE—the TPC and antiradical activity in the samples before digestion.

### 2.5. Statistical Analyses

The results are shown as the mean value $\pm$ standard deviation. Significant differences are considered as significant at $p \leq 0.05$ among the acquired samples and were determined by a *t*-test in the sensory tests. Analyses of variance (ANOVA) and Tukey's test is used to evaluate the effect of tested factors and to determine differences among the samples.

## 3. Results

### *3.1. Description of Sauerkraut Juice and Dehydrated Sauerkraut Juice*

#### 3.1.1. Total Phenols, Antiradical Activity and Bioaccessibility of the SJ and DSJ Samples

Total phenol content (TPC) is the highest in the sauerkraut juice before processing, as shown in Table 2. After the encapsulation process, in the dehydrated sauerkraut juice (DSJ), the TPC decreases by half, which could be explained by the high core-to-wall ratio, which helps to protect bioactive compounds. As scientists previously have confirmed that modified starch entraps the core material in the starch granule [27]. The antiradical activity is also affected by the optimum choice of spray-drying parameters that can retain the activity [6].

**Table 2.** Total phenol content and antiradical activity by ABTS$^+$ in sauerkraut juice (SJ) and dehydrated sauerkraut juice (DSJ).

| Parameters | SJ | DSJ |
|---|---|---|
| TPC, mg 100 g GAE, dw * | 713.7 $\pm$ 43.2 a ** | 359.5 $\pm$ 7.7 b |
| ABTS, mg TE 100$^{-1}$, dw | 15.50 $\pm$ 1.84 a | 28.62 $\pm$ 2.03 b |

* mg GAE–gallic acid equivalent, dry weight (dw). ** Values with different letters are significantly different ($p \leq 0.05$).

Bioaccessibility (BAC) is defined as a share of bioactive compounds that is released from the food matrix and become available for absorption after ingestion [28]. Bioaccessibility is calculated as equality of the analyses after the GIT against the analyses before GIT. The bioaccessibility is considered high if the BAC index is higher than 1 [17].

For both analyzed products–SJ and DSJ—the bioaccessibility for ABTS$^+$ exceeds 1.2 and is considered as high, as shown in Figure 1.

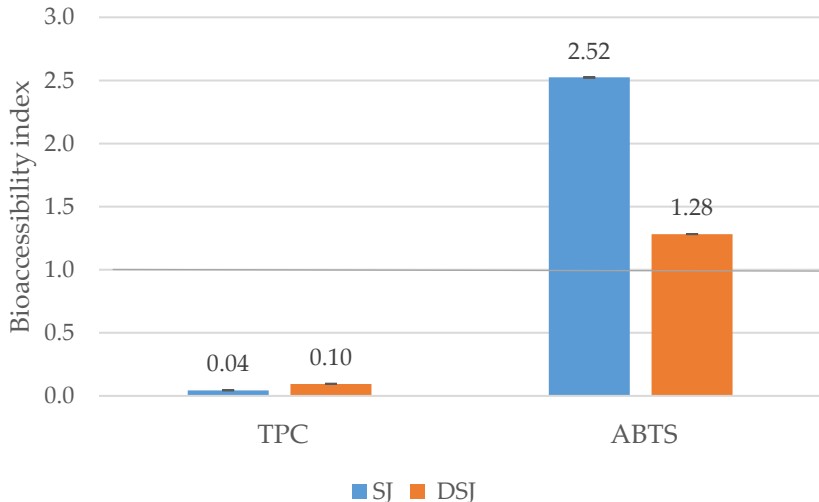

**Figure 1.** Bioaccessibility index (BAC) of the sauerkraut juice (SJ) and dehydrated sauerkraut juice (DSJ) based on TPC and ABTS$^+$ scavenging activity after in vitro digestion.

The bioaccessibility based on TPC is very low for SJ and DSJ and is below 0.2. This can be explained by the compound interaction with gastric juices and enzymes as well as the wall material or the combination of several, which plays an important role in the release of the phenolic compounds in the simulated gastrointestinal tract [18].

#### 3.1.2. Aromatic Volatiles in Sauerkraut Juice and Dehydrated Sauerkraut Juice

Profiles for aromatic volatiles was determined for sauerkraut juice and dehydrated sauerkraut juice and are presented in Figure 2.

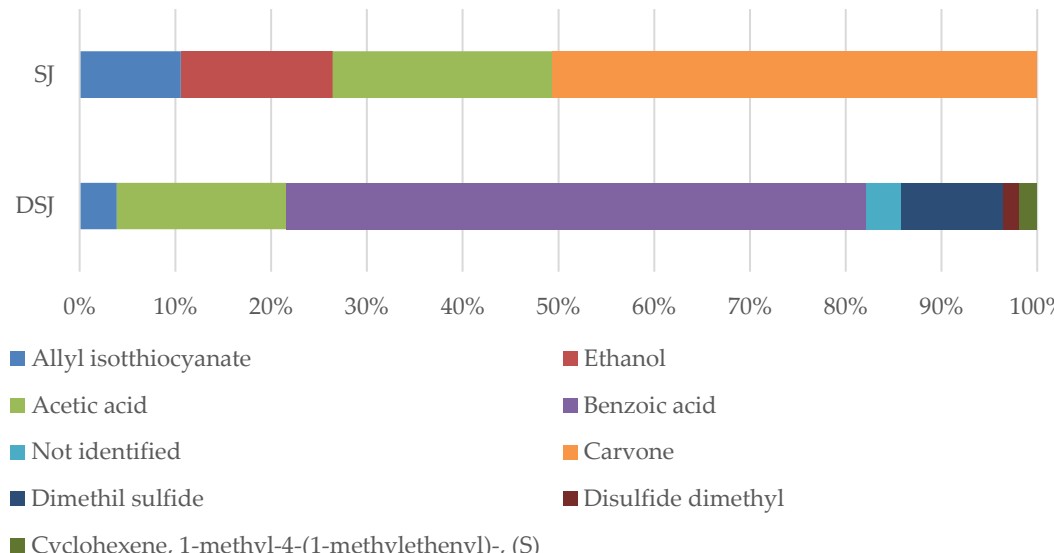

**Figure 2.** The percentage of volatile compound peak areas in sauerkraut and dehydrated sauerkraut juice.

There were two common compounds in sauerkraut juice before and after spray-drying: acetic acid with a sour taste, and allyl isotthiocyanate with a pungent taste of mustard, horseradish, and wasabi (mean values and standard deviations in Table S1, Supplementary Materials). The metabolized products of glucosinolates–isotthiocyanates are the prime sources of the characteristic flavor of the Brassica vegetables [3,28]. The highest peak area in sauerkraut juice was carvone with a caraway and spearmint-like odor, while for dehydrated sauerkraut juice it was benzoic acid with a faint, pleasant odor. Other compounds are characteristic to each of the raw material.

### 3.2. Bread Samples

3.2.1. Total Phenols, Antiradical Activity and Bioaccessibility of the Bread Samples

The aim of this experiment was to evaluate, if the addition of DSJ affects the bread quality, and can increase the bioactive compound content and the bioaccessibility in the wheat bread. The results, shown in Table 3, represent a significant influence on the total phenol content by the addition of DSJ to the wheat bread. The TPC content and antiradical activity by ABTS$^+$ in the Bread DSJ sample is higher by 66% and 56% accordingly. It has previously been reported that plant materials and food production by-products added to flour products increase the TPC and antiradical activity [29,30]. In our study, supplementing wheat bread with DSJ increases the TPC content in the sample, thus the food matrix provided suitable conditions for the release (or interaction) of the compounds.

**Table 3.** Total phenol content and antiradical activity by ABTS$^+$ in the bread samples.

| Parameters | Bread C | Bread DSJ |
|---|---|---|
| TPC, mg 100 g GAE, dw * | 54.36 ± 1.33 a ** | 82.56 ± 0.98 b |
| ABTS, mg TE 100$^-$, dw | 4.614 ± 0.241 a | 8.232 ± 0.563 b |

* mg GAE–gallic acid equivalent, dry weight (dw). ** Values with different letters are significantly different ($p \leq 0.05$).

Although the TPC was higher in the Bread DSJ, the bioaccessibility index BAC, shown in Figure 3. is higher in the Bread C–1.10 while in the DSJ sample it is 0.65. This suggests that wheat bread is rich in bioavailable phenolic compounds, however addition of DSJ may interact with other compounds to form indigestible compounds and do not protect the TPC from the severe environment of the stomach [31].

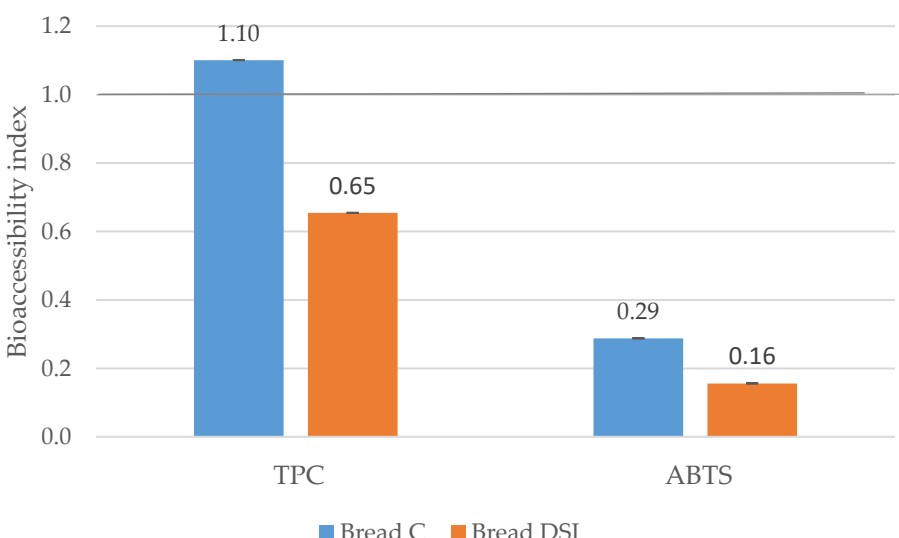

**Figure 3.** Bioaccessibility index (BAC) of the bread samples based on TPC and ABTS[+] scavenging activity after in vitro digestion.

That is contrary to the bread samples enriched with green coffee [17], where all of the samples were highly bioaccessible in vitro, but the control sample was higher than two. The interactions among the food matrix, phenolic compounds, and GIT enzymes is still under investigation [28].

### 3.2.2. Aromatic Volatiles in the Bread Samples

There were nine volatile compounds detected in the bread samples, and eight of them exceeded 5% and are shown in Figure 4. Hexanal did not exceed 1%. The highest peak area was for benzaldehyde, giving volatile oil-of-almond odor, and it was higher with the DSJ addition, being 29.15% and 33, 60%, accordingly (mean values and standard deviations in Table S2, Supplementary Materials). A very distinct nuance—a caraway-like odor–in the bread sample was detected with the DSJ addition. There is 1% of caraway added in the production process of sauerkraut in Ltd. "Dimdiņi" (Lizums, Latvia), and this volatile compound is so strong to remain through the spray-drying process and the bread baking. A freshly baked wheat bread, with no caraway or DSJ added, gives it a rose-like aroma.

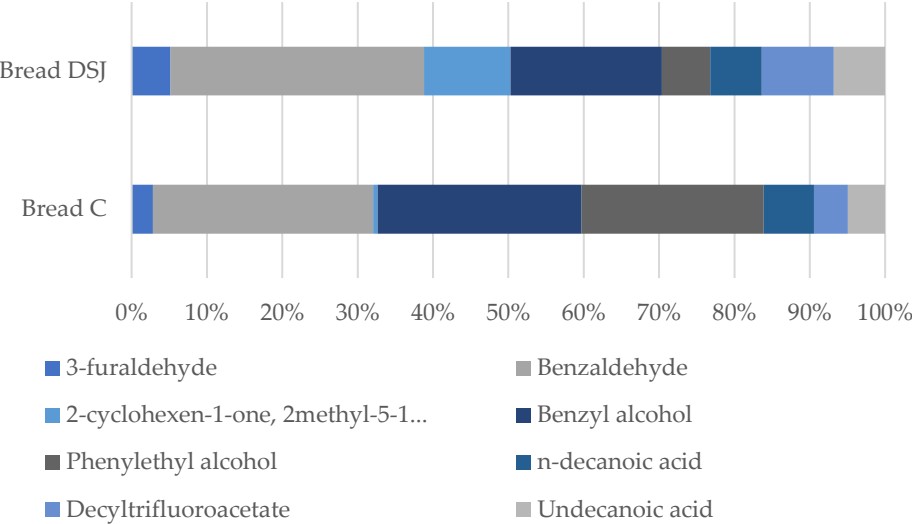

**Figure 4.** The percentage of volatile compound peak areas in the bread samples.

A sensory evaluation was carried out for bread with dehydrated sauerkraut juice. Fifty-two consumers were invited to rate overall liking, structure, taste, and aroma on a 5-point hedonic scale.

There are no significant differences in overall liking of Bread C and Bread DSJ; both samples are equally liked, and no significant differences are observed in structure, taste and aroma, as shown in Figure 5.

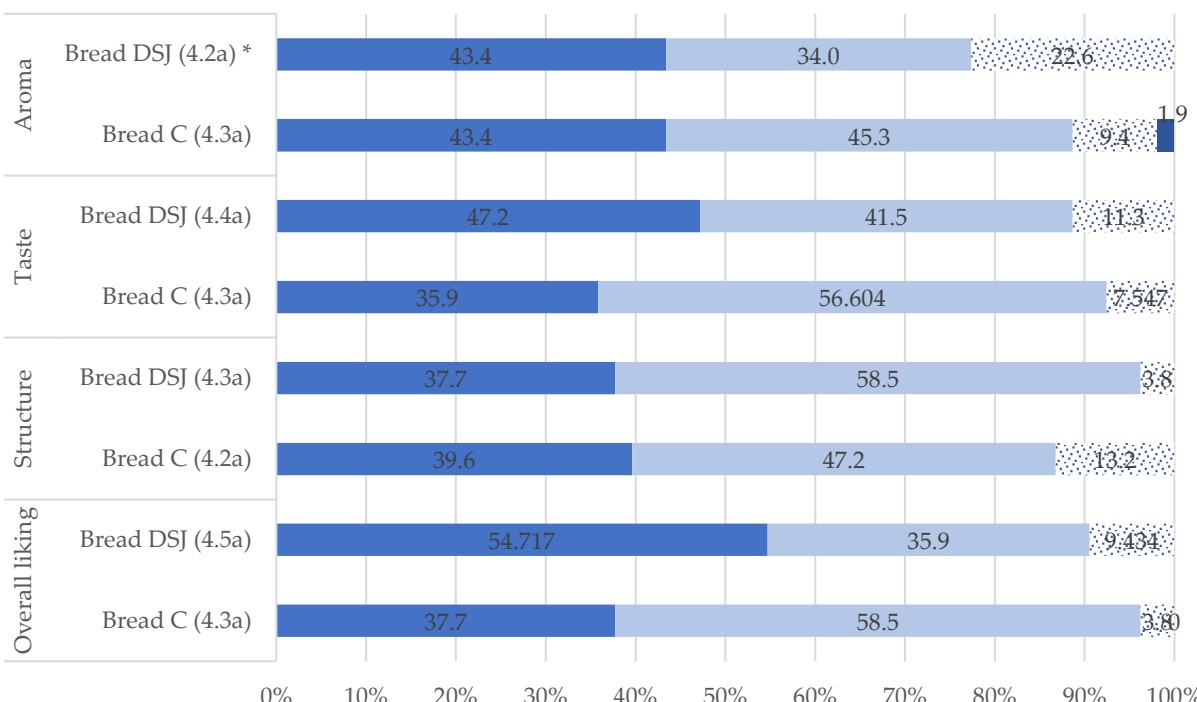

**Figure 5.** Hedonic evaluation of the bread samples. * Value in brackets present mean value of hedonic evaluation and the same letter along the values shows that the difference between the means between two samples is not statistically significant.

Other authors discussed that mean in some cases is not representative for hedonic evaluation due to differences in frequency of evaluations [32]. Therefore, Figure 5 presents the frequency of each evaluation in percentages and a few trends could be observed. It could be concluded that for overall liking and taste of Bread DSJ, more consumers selected evaluation "like extremely", 54.7% and 47.2%, respectively, and it is higher compared to Bread C. More people extremely like bread with DSJ. Additionally, it could be seen that aroma of Bread DSJ got 22.6% of evaluation–neither like nor dislike—showing that the aroma of this sample is not preferable for consumers in comparison to Bread C.

### 3.3. Meat Samples

3.3.1. Total Phenols, Antiradical Activity and Bioaccessibility of the Meat Samples

Mincing or grinding of meat influences total phenol content in meat products and its chemical composition consists of lipids, proteins, and polysaccharides that may interact with phenols and change their extractability from the samples [33]. In our study, there is a significant influence ($p < 0.01$) of DSJ addition in the meat samples on the TPC. The TPC content and antiradical activity by ABTS+ in the Meat DSJ sample is higher by 64% and 51%, accordingly, as shown in Table 4.

**Table 4.** Total phenol content and antiradical activity by ABTS$^+$ in the meat samples.

| Parameters | Meat C | Meat DSJ |
|---|---|---|
| TPC, mg 100 g GAE, dw * | 39.37 ± 1.62 a ** | 61.21 ± 1.03 b |
| ABTS, mg TE 100$^-$, dw | 2.924 ± 0.121 a | 5.721 ± 0.171 b |

* mg GAE–gallic acid equivalent, dry weight (dw). ** Values with different letters are significantly different ($p \leq 0.05$).

The bioaccessibility index for TPC of the meat samples is significantly higher than 1, as shown in Figure 6., and thus the compounds are available for absorption. The bioaccessibility of the control sample is significantly higher than the DSJ sample. The combination of proteins and phenolic compounds can affect the bioaccessibility of TPC, and is influenced by the specific compound interactions [34,35].

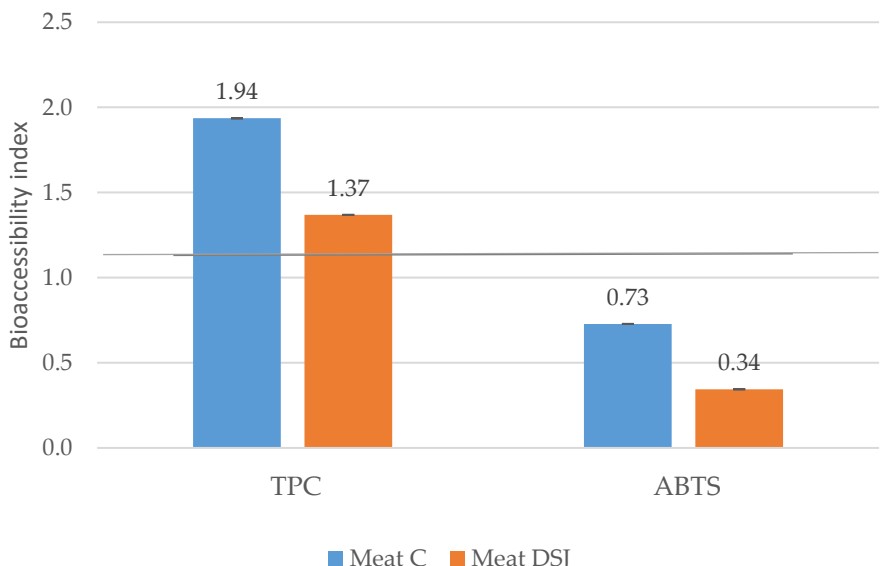

**Figure 6.** Bioaccessibility index (BAC) of the meat samples based on TPC and ABTS$^+$ compound activity after in vitro digestion.

As Nagar [36] has studied, dissolved oxygen levels in the intestinal phase and bile contribute to the decrease in bioaccessibility; the absence of oxygen increases the bioaccessibility of polyphenols.

Additionally, the bioaccessibility for ABTS$^+$ is higher in the control sample–0.73, while in the DSJ sample, it is 0.34, and thus the BAC of the antiradical activity is considered as low.

### 3.3.2. Aromatic Volatiles in the Meat Samples

There were eight volatile compounds detected in meat samples. Seven of them exceeded 5% and are shown in Figure 7. The highest peaks for the Meat C sample were hexanal, mostly formed by oxidation of linoleic acid [37], and 3-furaldehyde, giving a fruity, grass-, and almond-like odor, also containing the volatile oil-of-almond. The highest peaks in Meat DSJ sample were benzaldehyde, characterized by a roasted peanut aroma [37], and benzyl alcohol, giving a faint aromatic and volatile oil-of-almond aroma, as well as furfural with an almond-like odor, and n-decanoic acid with a rancid, unpleasant odor, mostly formed in the process of lipid hydrolysis and oxidation [37] (mean values and standard deviations in Table S2, Supplementary Materials). There is no caraway odor found in the meat sample.

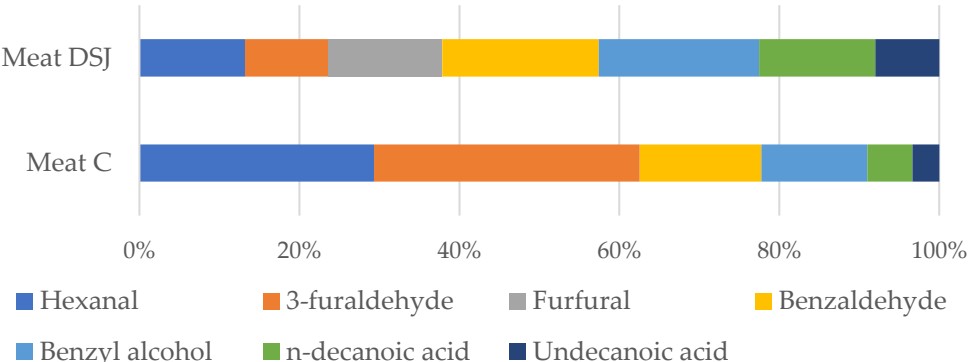

**Figure 7.** The percentage of volatile compound peak areas in the meat samples.

A sensory evaluation was carried out in the meat samples just as described above in the bread samples applying the Hedonic scale. Due to DSJ specific aroma and taste properties, it was useful to distinguish the effect and differences between the control and DSJ sample. In the bread samples, statistics showed no significant differences between samples, but for meat samples, a different trend was observed, as shown in Figure 8.

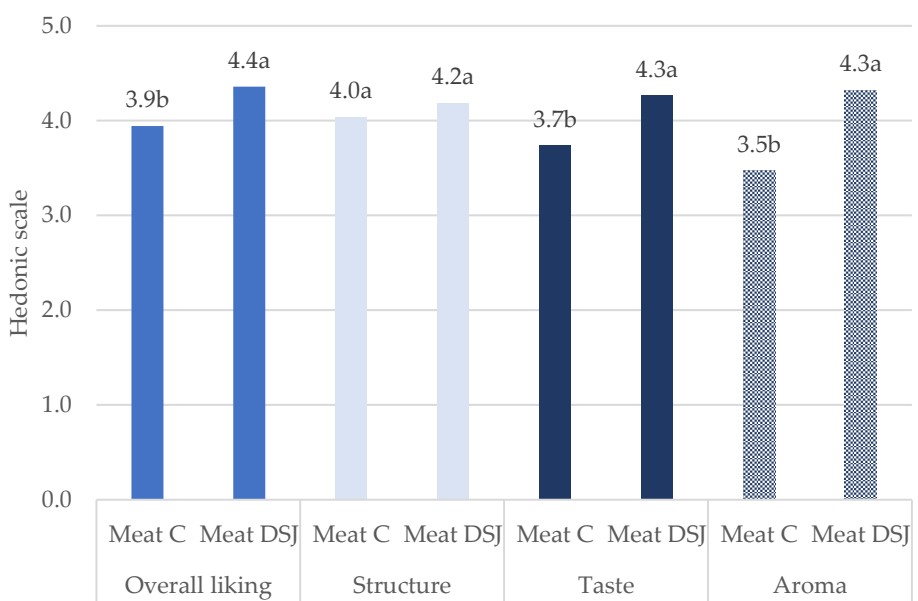

**Figure 8.** Hedonic evaluation of the meat samples. The same letter along the values shows that the difference between the means of two samples is not statistically significant.

No significant differences were in the structure ($p > 0.05$) of the samples whereas significant differences in the overall liking, taste, and aroma of the samples ($p < 0.05$) were determined and in all cases, higher evaluation was for a sample with added DSJ. It could be explained by traditional cooking and serving of meat with sauerkraut, and these tastes paired for Latvian consumers that the flavor was considered acceptable and highly evaluated. Therefore, meat products could be prospective for application of DSJ in food.

## 4. Discussion

Sauerkraut juice (SJ) and dehydrated sauerkraut juice (DSJ) has an ample amount of bioactive compounds, and contains phenolic compounds, vitamins, and minerals.

However, this study shows that the bioaccessibility of TPC is very low. There are numerous studies on phenolic compound bioaccessibility, mostly because the compounds are considered to have low-bioavailability [36], thus, the potential health benefits are under discussion [36,38]. The bioaccessibility of TPC is complex [28,36,39], since it is influenced

by many factors and depends on a diversity of plant polyphenols, the interaction of compounds, and demeanor in the digestive system [34,40] where the gut microbiota's hydrolytic activity can increase activity and bioavailability of polyphenols [41].

Sauerkraut juice and its products have a very distinct flavor and aroma that is composed of various volatile compounds, such as aldehydes, alcohols, sulfur compounds, esters, ketons, terpenes, furans, etc. [42], and can be metabolized from bioactive compounds, delivering health promoting attributes [15]. The main volatile compound of sauerkraut juice was carvone with a spearmint like odor and ethanol, both of them are characteristic to sauerkraut. The main volatile compound for DSJ was benzoic acid with a faint, pleasant odor. There were two common volatile compounds in SJ and DSJ–acetic acid with a sour odor, and allyl isothiocyanate with a pungent taste of mustard, horseradish, and wasabi. Allyl isothiocyanate, with antioxidant and antimicrobial properties, is one of the major volatile compounds of a specific cabbage flavor [43], however, the compound was not identified in the bread and meat samples tested in this study.

Dehydrated sauerkraut juice (DSJ) was tested in foods such as wheat bread and minced pork, and the effect of total phenol content (TPC) and antiradical activity by ABTS$^+$ in in vitro gastrointestinal digestion, and the compound bioaccessibility were evaluated.

The addition of DSJ significantly enriched wheat bread with TPC and antiradical activity, as it is shown in this study and is in agreement with previous observations of other authors, that adding vegetables with high phenol content to wheat bread can increase its total phenol content [44].

The bioaccessibility of TPC in the wheat-based food matrix is different than in meat-based products. Wieca [17] have confirmed that wheat flour contains bound phenolics that are easily released during simulated digestion. However, bioaccessibility is affected by the choice of wall material [45]. Dehydrated sauerkraut juice was acquired via spray-drying, and starch solution was used as a wall material. Encapsulated ascorbic acid in enzyme-hydrolyzed starch [27] and rutin in debranched lentil starch coating material is released in the intestinal digestion phase [46]. The choice and combination of wall material is crucial, and the desired release of the bioactive compounds and the interaction of the food matrix is applied to [35,45]. The addition of DSJ did not promote TPC bioaccessibility of bread and meat samples.

The volatile compounds of bread samples are characteristic of the product, the main compound being benzaldehyde with rose- and almond-like odor, ref. [47] yet the addition of supplements affects the profile. In our study, a caraway-like odor was detected in the Bread DSJ. Caraway is used in the production process of sauerkraut.

There were no significant differences in the sensory evaluation of structure, taste, aroma, and overall liking of the Bread samples C and DSJ. However, enriching wheat bread with lyophilized kale had a decrease in acceptability of taste and aroma, but it was influenced by individual preferences [8]. Additionally, adding broccoli leaf powder to gluten free sponge cakes increased their antioxidant capacity, but sensory quality was affected [48]. With this, it can be concluded, that the addition of DSJ to wheat bread does not affect sensory characteristics, but increases TPC and antiradical activity.

DSJ also significantly increased the TPC content and antiradical activity of the meat samples. Studies show, that incorporating plant-based additives in meat products, significantly increase their TPC [49–51], however, sensory profile is disputable. The bioaccessibility of TPC in the meat samples exceeded 1 and is considered as good.

Studies of meat protein and phenolics action in vitro are scarce to the best of our knowledge, but Rashidinejad [52] has investigated that milk proteins bind the phenols and can affect their release due to the interaction between them [35]. The TPC is affected differently in GIT, depending on the food matrix, its physicochemical characteristics, and its composition [39,51]. When testing poultry feed with the addition of lucerne or chicory, the total phenol content decreased in the gastric phase but increased in the intestinal phase [51]. Additionally, higher content of bioactive compounds in the food matrix does not always ensure the same results in and after the GIT [31], as well as the accessibility of

the compounds is influenced by numerous conditions such as the complexity of phenolic compounds in food matrix, the metabolic pathway, etc. [28,39].

Furthermore, Cantele [40] has concluded that the absorption of phenols from the solid food matrices is more challenging, as opposed to liquid matrices, because they first need to undergo mechanical, chemical, and enzymatic processes, whereas phenols from the liquid matrices are available straight away.

As Flores [53] investigated in their study about microencapsulated blueberry anthocyanins during in vitro with two types of wall material, whey protein and gum arabic, most of the whey protein microcapsules phenolic compounds are degraded in the intestinal digestion, whereas gum arabic, being a complex heteropolysaccharide, remains minimally digested. as Additionally, the stability of total phenol content is higher in the gastric phase, yet it decreases in the small intestine, because of bile [36] (especially phenolic acids), or the compounds transform into other compounds during digestion [28,38,54].

Benzaldehyde and 3-furaldehyde with almond-like odors are dominant in the meat samples, yet the addition of DSJ amplifies the rancid, fatty odor with n-decanoic acid. Volatile compounds in meat are affected by cooking time and temperature and different reactions during the process (such as Maillard reaction) and a series of nutrient degradation [37].

In the sensory evaluation, there were significant differences in overall liking, taste, and aroma of the meat samples. The preference was for the Meat DSJ sample, and can be explained by the additional flavor and salty taste. However, adding dried cabbage to minced mutton patties caused a decline in the overall liking, flavor, and texture [49].

In conclusion, DSJ could potentially be used as a supplement in food applications, to enrich their functional properties, if it is acceptable for taste and aroma sensory attributes. Research on sauerkraut juice and its production products is scarce and further research is recommended.

**Supplementary Materials:** The following supporting information can be downloaded at: https://www.mdpi.com/article/10.3390/app13053358/s1, Table S1: Relative area of volatile compounds in SJ and DSJ; Table S2: Relative area of volatile compounds in bread and meat samples.

**Author Contributions:** Conceptualization, L.J. and Z.K.; methodology, Z.K. and K.M.; formal analysis, L.J. and Z.K.; investigation, L.J.; resources, S.K.; data curation, Z.K.; writing—original draft preparation, L.J.; writing—review and editing, Z.K. and S.K.; supervision, Z.K. All authors have read and agreed to the published version of the manuscript.

**Funding:** This research was funded by ESF project grant number 8.2.2.0/20/I/001/TOPIC ES32.

**Institutional Review Board Statement:** Not applicable.

**Informed Consent Statement:** Informed consent was obtained from all subjects involved in the study.

**Data Availability Statement:** Not applicable.

**Acknowledgments:** This research was financially supported by the ESF project Nr. 8.2.2.0/20/I/001/ TOPIC ES32—"Latvia University of Life Sciences and Technologies transition to the new doctoral funding model" and the European Innovation Partnership for Agricultural Productivity and Sustainability Working Group Cooperation project 18-00-A01612-000020. Z.K.; S.K. and K.M. acknowledges the support.

**Conflicts of Interest:** The authors declare no conflict of interest.

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
