# Peer review of "Dehydrated Sauerkraut Juice in Bread and Meat Applications and Bioaccessibility of Total Phenol Compounds after In Vitro Gastrointestinal Digestion"

_applsci, doi:10.3390/app13053358_

Round 1

Reviewer 1 Report

The topic of this research is within the current food trends such as the search for alternatives to improve the nutritional value of food matrices such as bread and meat, and it is interesting that these bioactive compounds come from a source such as sauerkraut, which not commonly used in these products.  However, the presentation of the research has problems in terms of the description of the knowledge gap it seeks to solve, the description of the methodology presents some gaps that hinder its reproducibility, the presentation of the tables and figures makes it difficult to understand the data obtained in each experiment, and there is no clear discussion of the results. Therefore, I consider that the manuscript should be rewritten, and to facilitate the process, I would like to indicate some suggestions:

·         Lines 12- 13 The information in these lines is related to results or its information as the introduction.

·         Line 16. was higher compared to which samples? and which samples are control?

·         Rewrite the introduction because it is not clear the problem and the knowledge gap to be solved, on the contrary, other topics are indicated without further contribution, for example, some research on DSJ and applications in matrices are mentioned, which are not clearly associated with the research presented in the manuscript. In addition, there is information associated with methodology. Likewise, check the writing of the aim.

·         The description of the process to obtain the samples with DSJ is not clear. 

·         The process of obtaining samples (bread and sausages) with DSJ does not allow their reproducibility, so it is necessary to describe these processes more clearly.

·         Line 125 - Check the writing of the folin reagent

·         Why was the DPPH measurement included?

·         It is important to include a description of the slight modifications made in the assays

·         The participants are either trained individuals or consumers. To improve the way in which participants are described in the sensory evaluation panel 

·         ¿What was the size of the samples? i.e. the bread and sausages were cut into small pieces or is it only a 30 g sample for Static in vitro digestion

·         The presentation and analysis of the results are very poor, there should be greater depth in the discussion of the results with other authors, and in turn, show the contribution of the study.

·         There are writing problems, which makes it difficult to understand the text. 

·         The presentation of the tables is not in accordance with the author's instructions given by the journal.

·         The data shown in table 3 do not correspond to the bread and sausage samples.

·         The references are not written correctly; the author's guide should be revised.

Author Response

Dear Rewiever,

Thank you for your time and input in our study! Your remarks and comments are appreciated and taken into account. 

Best regards!

Reviewer 2 Report

This manuscript entitled “Dehydrated sauerkraut juice in bread and meat applications and bioaccessibility of total phenol compounds after in vitro gastrointestinal digestion.” is an interesting and original study.

The paper is clearly presented and results are very useful. However, I have some suggestions:

1.     Line 78: Please rephrase the aim.

2.     Line 145: Were the sensory participants trained, were they consumers, is the number of participants determined by any regulations to be representative?

3.     Line 173: Improve the expression of the equation

4.     Figures: Please add the standard deviations or standard errors in the figures.

5.     Tables. The number of digits of the error value depends on the place where the significant digit appears and the number of digits of the corresponding data should be adjusted by taking into account the corresponding error values. In this way, each value in the Tables must be expressed with the significant digits according to the significant digits of each error value (in this case the significant digits of the standard deviation). Please correct it.

6.     Improve the discussion, it is poor. Needs more relation and explanation of what is going on.

Author Response

(The authors gave the same response as above.)

Round 2

Reviewer 1 Report

I considered that no all observations indicated was included in this new manuscript, however, new version shows a good discussion of the results.

Author Response

Thank you for valuable suggestions that let us improve our work and thank you for your time and input delving into our manuscript.

Sincerily, Liene Jansone

Reviewer 2 Report

The new version of the manuscript is much improved.

Author Response

(The authors gave the same response as above.)
